

# A model for rapid wildfire smoke exposure estimates using routinely-available data - rapidfire v0.1.3

Sean Raffuse[1], Susan O'Neill[2], and Rebecca Schmidt[3]

[1]Air Quality Research Center, University of California Davis, Davis, CA, United States
[2]Pacific Northwest Research Station, USDA Forest Service, Seattle, WA, United States
[3]Department of Public Health Sciences, MIND Institute, University of California Davis School of Medicine, Davis, CA, United States

**Correspondence:** Sean Raffuse (sraffuse@ucdavis.edu)

**Abstract.** Urban smoke exposure events from large wildfires have become increasingly common in California and throughout the western United States. The ability to study the impacts of high smoke aerosol exposures from these events on the public is limited by the availability of high-quality, spatially-resolved estimates of aerosol concentrations. Methods for assigning aerosol exposure often employ multiple data sets that are time consuming and expensive to create and difficult to reproduce.

As these events have gone from occasional to nearly annual in frequency, the need for rapid smoke exposure assessments has increased. The rapidfire R package (version 0.1.3) provides a suite of tools for developing exposure assignments using data sets that are routinely generated and publicly available within a month of the event. Specifically, rapidfire harvests official air quality monitoring, satellite observations, meteorological modeling, operational predictive smoke modeling, and low-cost sensor networks. A machine learning approach (random forests regression) is used to fuse the different data sets. Using rapidfire,

we produced estimates of ground-level 24-hour average particulate matter for several large wildfire smoke events in California from 2017-2021. These estimates show excellent agreement with independent measures from filter-based networks.

## 1 Introduction

Changes in climate in the western United States, and elsewhere, are driving larger, more intense fires with greater smoke impacts on larger populations (Burke et al., 2021), and these trends are projected to continue (Hurteau et al., 2014). The wildfire

seasons of 2020 and 2021 produced some of the highest concentrations of particulate matter less than 2.5 microns in diameter ($PM_{2.5}$) ever observed in monitoring stations around California, some for several days or weeks. Despite reductions in ambient $PM_{2.5}$ driven by air pollution regulations, areas of the western United States are seeing increasing concentrations (McClure and Jaffe, 2018).

There are widespread concerns about potential health consequences of wildfire exposures on vulnerable populations as the

increasingly reach populated areas. From 2008-2012, it was estimated that over 10 million individuals in the US experienced unhealthly air quality levels (average daily fire-$PM_{2.5} > 35\ \mu g\ m^{-3}$) associated with exposure to wildfire for more than 10 days (Rappold et al., 2017). This number is expected to have risen several-fold in the decade since given the increase in wildfire events across the world. Additionally, long-range transport of wildfire PM has been associated with adverse health effects in



susceptible populations thousands of miles away (Kollanus et al., 2016, Le et al. (2014)).

Wildfire smoke is associated with premature deaths (Johnston et al., 2012, Chen et al. (2021a)), and significant cardiovascular (Chen et al., 2021b) and respiratory morbidity (Reid et al., 2016), including asthma exacerbations. Certain subpopulations are more susceptible to the health impacts of air pollution and wildfire smoke, including the elderly, pregnant women, and those with underlying health conditions such at asthma (Chen et al., 2021b). Few studies have examined long-term health outcomes in relation to chronic exposures to high concentrations of wildfire smoke. Prenatal wildfire smoke exposure has been linked to

adverse birth outcomes, including preterm birth (Heft-Neal et al., 2022), and lower birth weight (Holstius et al., 2012, Abdo et al. (2019)), especially with exposure in the second or third trimester. In contrast to studies of ambient air pollution, associations between wildfire smoke and adverse birth outcomes did not differ by race, ethnicity, or income, but differed by baseline smoke exposure. Many epidemiologic studies have linked early life air pollution exposure to increased autism spectrum disorder risk (Volk et al., 2011, Volk et al. (2013), Dutheil et al. (2021)) and to cognitive functioning impairments (Loftus et al.,

2020, Clifford et al. (2016), Loftus et al. (2019), Chiu et al. (2016)).

Evidence suggests that wildfire PM could induce higher toxicity than other ambient air PM (Wegesser et al., 2009, Kim et al. (2018), Wegesser et al. (2010), Franzi et al. (2011)) and is associated with about 10 times higher increase in hospital admissions for respiratory health than PM from other sources (Aguilera et al., 2021a), including in young children (Aguilera et al., 2021b). With climate predictions for increased occurrence and severity of wildfires, there is a growing need to understand

which populations are at highest risk and PM concentrations of concern to inform adverse health mitigation strategies. Yet, many gaps remain in our understanding of the linkages between wildfire smoke and human health (Black et al., 2017). A critical challenge is in characterizing personal or population exposures during high-intensity events. There are many methods for estimating exposure to ambient pollution, including spatial interpolation of measured values, chemical transport modeling, remote sensing, land-use regression modeling, data fusion and machine learning, and combinations of all of these approaches

(e.g., Reid et al. (2015), Zhang et al. (2020), Al-Hamdan et al. (2014), Cleland et al. (2020), Hoek et al. (2008)). The rapidly changing conditions during wildfire smoke events can confound otherwise high-performing approaches (O'Neill et al., 2021). There are several barriers to the adoption of existing methods for exposure assignment. These can include data availability for the study location, data latency, and high-performance computing requirements. The combination of increasing frequency of smoke events and the proliferation of smoke exposure human health studies drives a need for exposure modeling that is quick

and inexpensive.

There has been a rapid proliferation of low-cost sensors for air quality within the past decade. While these sensors do not measure $PM_{2.5}$ with the same fidelity as the regulatory monitoring conducted by federal and local air quality agencies, they represent a new resource for $PM_{2.5}$ assessment with relatively dense spatial coverage. Many low-cost PM sensors operate with similar principles, using a laser to count particles that scatter light in the optical range, with sensitivities peaking for aerosols

with median scattering diameter < 0.3 μm (Ouimette et al., 2022). Recent studies have shown the value of incorporating low-cost sensor networks into $PM_{2.5}$ exposure modeling (Bi et al., 2020).

Past work has shown that a data fusion approach that combines ground-based air quality monitors, transport modeling that incorporates wildfire emissions, satellite observations, and meteorological variables can be effective in predicting $PM_{2.5}$ expo-

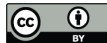



sure during large wildfire events (Zou et al., 2019, and O'Neill et al. (2021)) and prescribed fires (Huang et al., 2021).

We developed methods and a suite of tools for rapidly predicting PM$_{2.5}$ exposure, particularly during wildfire smoke events, using readily available data with low latency (less than one month). The tools are contained within a package written in the R programming language called rapidfire (relatively accurate particulate information derived from inputs retrieved easily). rapidfire adapts and builds upon the methods of Zou et al. (2019) and O'Neill et al. (2021), replacing retrospective chemical transport modeling and other data sets developed for research with smoke forecast modeling and "off-the-shelf" data sets that

are routinely available and easily acquired. A major addition is the incorporation of low-cost sensor data. This paper describes the data sets and algorithms used in the rapidfire package and presents an example case study during five recent extreme wildfire seasons in California.

## 2 Methods

In this study, datasets and algorithms are applied to time periods of large California wildfires from 2017-2021. Table 1 summa-

rizes some of the major California wildfires and the area burned for the year. Figure 1 shows the wildfire locations, as detailed by the California Department of Forestry and Fire Protection's Fire and Resource Assessment Program (FRAP). Extreme fire weather conditions fueled the October 2017 wine country wildfires in the Napa and Sonoma counties of central coastal California (Mass and Ovens (2019)) and over 7 million people were impacted by unhealthy levels of smoke (O'Neill et al. (2021)). 2018 began in July with wildfires such as the Carr, Ferguson, and Mendocino Complex (Mueller et al. (2020)) and extended

through November with the Camp (Mass and Ovens (2021)) and Woolsey wildfires. 2019 was a relatively low activity fire year in comparison, but the Kincade wildfire again impacted the wine country in Oct-Nov. The 2020 wildfire season was relatively quiet until the middle of August when widespread lightning ignited many wildfires across central and northern California, including the coastal range south of San Francisco. 2021 burned about two-thirds the acres as in 2020, but over a longer duration, starting about a month earlier in July. These different patterns and level of smoke impacts are seen in Figure 2 which shows

24-hour average PM2.5 concentrations from permanent and temporary monitors across the state of California and satellite imagery of the smoke and satellite hot spot detections.

**Table 1.** Modeled time periods and major Northern California wildfires. Annual area burned is from the US National Interagency Fire Center (NIFC; https://www.nifc.gov/fire-information/statistics)

| Year | Time Period | Major Fires | Annual Area Burned (ha) |
|------|-------------|-------------|------------------------:|
| 2017 | October | Atlas, Nuns, Pocket, Redwood Valley, Tubbs | 512,522 |
| 2018 | July 15 - September 15; November | Carr, Mendocino, Ferguson, Camp, Woolsey | 737,804 |
| 2019 | October 15 - November 15 | Kincade | 104,873 |
| 2020 | August - October | August, Creek, LNU Lightning, North, SCU Lightning | 1,656,035 |
| 2021 | August - October | Antelope, Caldor, Dixie, Monument, River | 903,933 |





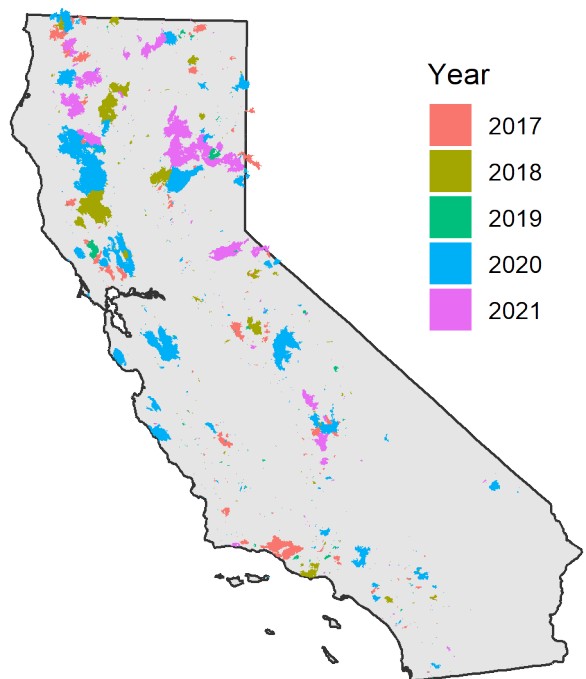

**Figure 1.** Locations of burned areas in California, 2017-2021.

## 2.1 Input Data Sets

Input data for rapidfire consist of ground-based monitors from three sources, aerosol optical depth from satellite instruments, and modeled meteorological and air quality data. Table 2 summarizes these data sources and the rapidfire functions used to access them and/or the location where the data can be obtained.

**Table 2.** Data sources used in rapidfire and the rapidfire function to access them or the location where sample data are available.

| Data Source | rapidfire function or location where available |
| --- | --- |
| AirNow Permanent PM2.5 Monitoring Data | rapidfire::get_airnow_daterange |
| IWFAQRP Temporary PM2.5 Monitoring Data | rapidfire::get_airsis_daterange |
| PurpleAir Air Sensor Data | rapidfire::pa_sensor_history, rapidfire::openaq_get_averages |
| MAIAC Aerosol Optical Depth | rapidfire::maiac_download |
| Example Smoke modeling data | DOI:10.5281/zenodo.7942846 |
| North American Regional Analysis (NARR) Meteorology | rapidfire::get_narr |





**Figure 2.** Temporal and area views of smoke impacts across California. Panels on the left show 24-hour PM2.5 concentrations from permanent and temporary monitors in California for July – November for 2017-2021. Data are color-coded by air quality index. Panels on the right show visible satellite imagery of smoke and satellite fire hot spot detections across California from NASA Worldview for October 13, 2017 during the wine country wildfires; November 9, 2018 during the Camp and Woolsey wildfires; October 27, 2019 during the Kincade wildfire; September 9, 2020 after widespread lightning ignition of wildfires in northern and central California; and August 19, 2021 when many wildfires were burning in northern California and the Sierras.





### 2.1.1 Permanent and Temporary Air Quality Monitoring Data

Hourly PM$_{2.5}$ observations are available from monitoring stations across the United States via the AirNow program, which is a partnership of the U.S. Environmental Protection Agency (EPA), National Oceanic and Atmospheric Administration, National Park Service, NASA, Centers for Disease Control, and tribal, state, and local air quality agencies (https://www.airnow.gov/).

Within California, about 117-141 monitors were operating during the study period. These permanent monitors are a mixture of federal reference method or federal equivalent method instruments; instruments of sufficient quality such that the data are used by EPA to determine attainment and non-attainment of the National Ambient Air Quality Standards (NAAQS).

During wildfires, temporary monitors are also deployed by the Interagency Wildland Fire Air Quality Response Program (IWFAQRP, (Congress.gov, 2019)) and the California Air Resources Board (CARB). These monitors are Environmental Beta

Attenuation Monitors (EBAM; Met One Instruments, Inc.). As discussed in O'Neill et al. (2021), laboratory (Trent 2006) and field (Schweizer et al., 2016) studies evaluating EBAM performance with federal reference method monitors found correlations greater than 0.9 with a tendency of the EBAM to overestimate PM2.5 especially when relative humidity was greater than 40% (Schweizer et al., 2016). Though not as accurate as the AirNow monitors, they are deployed in regions where smoke impacts are significant and permanent monitoring is sparse or absent. The locations of permanent and temporary monitors as of September

1, 2021 are shown in Figure 3a. The permanent monitors are concentrated in the coastal and valley regions where larger populations of people are located, while temporary monitors are focused in areas of complex terrain where most wildfires and smaller communities without air quality monitoring data are located.

Hourly PM$_{2.5}$ concentrations from both the permanent and temporary monitors were acquired using the `rapidfire::` `get_airnow_daterange` and `rapidfire::` `get_airsis_daterange` functions. These wrap the `monitor_subset`

function from the `PWFSLSmoke` R package [Mazama Science]. `rapidfire::` `recast_monitors` was then used to calculate daily 24-hr averages from the hourly data. At least 16 hours are required to produce an average. The daily average data from both the permanent and temporary monitors were combined into a single data set. 30% of this monitor data set was withheld for development and evaluation of the rapidfire model results. The remaining 70% were used to develop model variograms using `rapidfire::` `create_airnow_variograms`. These PM$_{2.5}$ observations were then log-transformed

and interpolated to estimate concentrations at locations away from the monitors using ordinary kriging (Wackernagel, 1995), providing a spatially complete dataset for use in the rapidfire data fusion.

### 2.1.2 Low-cost Sensors

There has been a proliferation of low-cost sensors that estimate PM$_{2.5}$ deployed by the public across the world in the last decade. We used data from the PurpleAir network, which has grown to over 6500 outdoor sensors in California as of 2021. Figure 1b

shows the locations of PurpleAir sensors reporting data on September 1, 2021. Coverage in populated areas is extensive.

While PurpleAir estimates of PM$_{2.5}$ concentration have been shown to be biased, and are dependent on humidity and aerosol type (Barkjohn et al., 2021), they still correlate with PM$_{2.5}$ observed at FEM monitors and provide invaluable spatial and



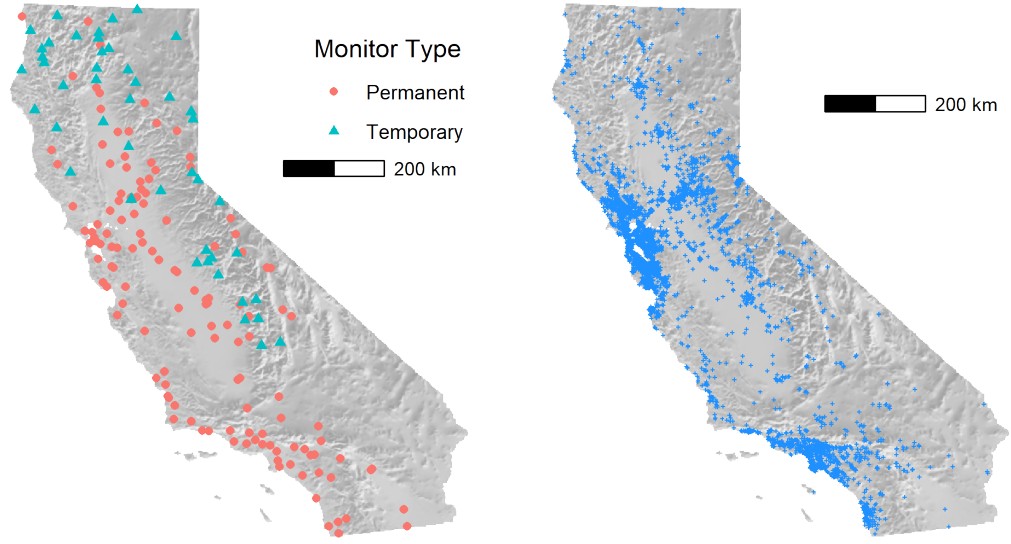

**Figure 3.** Map of permanent and temporary California monitor locations (left) and PurpleAir outdoor sensor locations (right); September 1, 2021.

temporal information that is not available with the relatively sparse network of monitors. Because these sensors are not quality controlled or validated, and their siting may be suspect, care must be taken when using them in modeling.

For time periods since February 2021, rapidfire acquires PurpleAir archive data using the OpenAQ application program-ming interface (API). OpenAQ is data platform that aggregates air quality data from around the world (OpenAQ, 2023). `rapidfire::openaq_find_sites` is first run to find all sensors within a specified geographic boundary. Then, `rapidfire::openaq_get_averages` can be used to download data for those sensors over the specified time period. At the time of publication, PurpleAir data from prior to February 2021 were not available via OpenAQ. For earlier time periods, rapidfire

queries data directly from the PurpleAir API. `rapidfire::pa_find_sensors` is used for finding all available outdoor PurpleAir sensors within a geographic bounding box. Then, `rapidfire::pa_sensor_history` can be run to acquire hourly PM$_{2.5}$ concentration estimates from each sensor. We employ a spatial test to remove sensors that are significantly differ-ent from their neighbors. `rapidfire::purpleair_clean_spatial_outliers` removes any sensors that are more that two standard deviations away from the median of all sites within 10km. PurpleAir estimates used in data fusion were

log-transformed and then interpolated using ordinary kriging.



### 2.1.3    Satellite Aerosol Optical Depth

Satellite aerosol optical depth (AOD) is a measure of the total columnar aerosol light extinction from the satellite sensor to the ground. AOD is indirectly related to $PM_{2.5}$, with the relationship depending on aerosol type, humidity, and aerosol vertical profile (Li et al., 2015). We used AOD from the Multi-Angle Implementation of Atmospheric Correction (MAIAC)

project (Lyapustin et al., 2011). MAIAC is an advanced algorithm that uses time series analysis and additional processing to improve aerosol retrievals, atmospheric correction, and, importantly, cloud detection from the MODerate-resolution Imaging Spectroradiometers (MODIS) onboard NASA's Terra and Aqua satellites. Past work has shown that thick smoke is often mistaken for clouds in the standard MODIS algorithms (van Donkelaar et al., 2011), which hampers their use in wildfire conditions.

The `rapidfire::maiac_download` function can be used to acquire the 1-km daily atmosphere product (MCD19A2) which contains AOD. Clouds prevent the retrieval of AOD, and there are sometimes clouds present even in the hot, dry conditions during California wildfires. The data fusion algorithm requires a complete data set, so a placeholder value must be used to gap-fill in locations under clouds. Previous work has used model-simuluated AOD, along with meteorological variables in a data fusion approach to gap-fill satellite-observed AOD (Zou et al., 2019). For this work, where clouds cover

less of the domain, we took a simpler approach. Missing AOD values were filled using a three-stage focal average, available in `rapidfire::maiac_fill_gaps_complete`, and illustrated in Figure 4. In the first stage, a focal mean of a 5-by-5 pixel square (5 km) is used. In the second stage, the window is increased to 9-by-9 and to 25-by-25 in the final stage. Any values that are still missing after the final stage are filled with the median value for the entire scene.

### 2.1.4    Smoke Modeling

Air quality models provide near-surface estimates of $PM_{2.5}$ on an output grid. We processed daily average $PM_{2.5}$ concentration values acquired from the BlueSky smoke prediction system (Larkin et al., 2009) developed by the USDA Forest Service (USFS) which first became operational in 2002 and has undergone significant development in recent years. The USFS runs over 30 simulations a day predicting near-surface 1-hr average $PM_{2.5}$ concentrations from wildland fire across the US at a variety of spatial extents and resolutions using the HYSPLIT dispersion model (Stein et al., 2015). For this work we extracted BlueSky

data from the California and Nevada Smoke and Air Committee (CANSAC; https://cansac.dri.edu/) domain that encompass California and Nevada for the months of July-November, years 2017-2021. In 2018 and 2019 the domain was at a 2-km resolution, and for 2019-2021 the domain was at a 1.33-km resolution. On some days, the model did not run successfully. For those days, data were backfilled by using the second or third day of a previous day's 72-hr model run. We chose this air quality dataset because it is available operationally, is of a high spatial resolution, and is focused specifically on modeling smoke

aerosols from wildland fires; however, other air quality modeling could be substituted.

Smoke prediction systems need to make many more assumptions than retrospective analyses and these assumptions, such as vegetation type and fuel loading, fire size and behavior, persistence of fire activity into the future, and using a meteorological forecast all have considerable implications for the quantity of emissions released from fires, and how those emissions



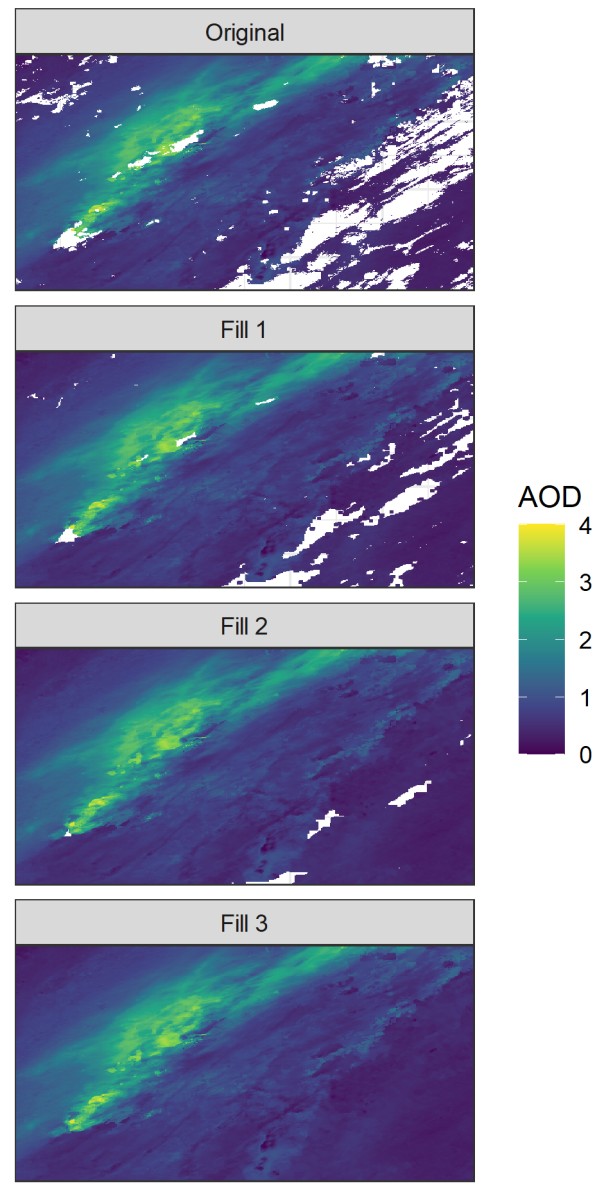

**Figure 4.** Illustration of MAIAC AOD gap filling

transport and undergo chemical reactions in the atmosphere (O'Neill et al., 2022, Kennedy et al. (2020) , Larkin et al. (2012)).
These assumptions and associated uncertainties can result in orders of magnitude spread in the estimated downwind PM2.5 concentrations (Li et al., 2020). Despite these issues, these systems are useful in providing information about potential smoke impacts (Lahm and Larkin, 2020), and the data are more available and can provide the underlying consistent dataset necessary to represent near-surface PM2.5 concentrations for successful applications of machine learning and health impact analyses.



Further, retrospective studies are not routinely available for long-term time periods (5-10 years or more) and maturing air quality forecasting systems when coupled with machine learning approaches such as provided here can provide the consistent high-quality datasets needed for health impact analyses.

### 2.1.5 Meteorology

Meteorological conditions can help explain the relationships between our inputs and observed $PM_{2.5}$. For example, the PurpleAir sensor is sensitive to relative humidity. AOD is sensitive to humidity and planetary boundary layer height. Following Zou et al. (2019), we included several meteorological variables in our model, including daily average temperature, winds, humidity, boundary layer height, and daily rainfall. These variables were acquired from the North American Regional Reanalysis (NARR) data set (Mesinger et al., 2006).

### 2.2 Data Fusion

We developed event specific models using random forests regression (RF). RF is a technique that uses a large number of randomly generated regression trees (Breiman, 2001). Each tree is constructed using a random subset of the training data and each node uses a random subset of the potential predictive variables. New values are estimated as the mean prediction of the individual trees. For each RF run, 500 trees were grown. A single tuning parameter, the number of variables selected at each node, was varied between 2 and 5. The model was trained using 10-fold cross-validation. Internally, `rapidfire::develop_model` uses the randomForest R package.

For the final model, 10 predictor variables were used (Table 3). $PM_{2.5}$ from the monitors was used as both a predictor and a target variable. A random subset of 30% of the monitoring data was withheld for model validation. Given a list of locations and dates, the final result from `rapidfire:: predict_locs` is a table with the 10 input variables plus the resulting modeled $PM_{2.5}$ for each location and date.

**Table 3.** Predictor variables used in the rapidfire RF model.

| Variable | Name | Description | Units |
|---|---|---|---|
| PM25_log_ANK | Monitors | Log-transformed, interpolated $PM_{2.5}$ from permanent and temporary monitors | $\mu g\ m^{-3}$ |
| PM25_log_PAK | PurpleAir | Log-transformed, interpolated $PM_{2.5}$ estimates from PurpleAir sensors | $\mu g\ m^{-3}$ |
| PM25_bluesky | BlueSky | Daily average ground-level $PM_{2.5}$ predictions from BlueSky smoke model | $\mu g\ m^{-3}$ |
| MAIAC_AOD | AOD | Gap-filled daily AOD from MAIAC | unitless |
| air.2m | Temperature | Daily average ambient temperature at 2m above ground level from NARR | $K$ |
| uwnd.10m | Wind u | Daily average u component of wind at 10m above ground level from NARR | $m\ s^{-1}$ |
| vwnd.10m | Wind v | Daily average v component of wind at 10m above ground level from NARR | $m\ s^{-1}$ |
| rhum.2m | Humidity | Daily average relative humidity at 2m above ground level from NARR | $\%$ |
| apcp | Precipitation | Daily total precipitation amount from NARR | $cm$ |
| hpbl | PBL Height | Daily average height of the planetary boundary layer from NARR | $m$ |



# 3 Results and Discussion

## 3.1 Model Evaluation and Comparison with Measurements

To demonstrate the performance of the rapidfire system we developed models for five large wildfire smoke events from 2017-2021 in Northern California (Table 1) and evaluated the modeling against two data sets of PM$_{2.5}$ observations, 1) the permanent and temporary hourly monitors described above, and 2) 24-hr filter-based measurements from the Interagency Monitoring of PROtected Visual Environments (IMPROVE) network and Chemical Speciation Network (CSN). Six quantitative analysis metrics are used to evaluate model performance (Table 4).

**Table 4.** Definitions of quantitative analysis metrics.

| Metric | Equation |
|---|---|
| $r^2$ | $\dfrac{\sum_i (\hat{Y}_i - \bar{Y})^2}{\sum_i (Y_i - \bar{Y})^2}$ |
| Root Mean Square Error (RMSE) | $\sqrt{1 - r^2} SD_Y$ |
| Median Bias | $med(\hat{Y}_i - Y_i)$ |
| Normalized Bias (%) | $100 * med(\dfrac{\hat{Y}_i - Y_i}{Y_i})$ |
| Median Error | $med(\dfrac{\hat{Y}_i - Y_i}{Y_i})$ |
| Normalized Error (%) | $100 * med(abs(\dfrac{\hat{Y}_i - Y_i}{Y_i}))$ |

Model predicted PM$_{2.5}$ values were compared against withheld measurements from the permanent and temporary monitoring networks using rapidfire and three other modeling techniques: 1) ordinary kriging (OK) interpolation of AirNow monitors, 2) OK interpolation of PurpleAir sensors, and 3) multiple linear regression (MLR) using the same inputs as those used for the rapidfire modeling. California is a state with a relatively greater number of surface air quality monitors than many other states in the western US. Thus we wanted to investigate the performance of rapidfire data fusion results together with the simpler OK and MLR models. Comparative model performance metrics are presented in Table 5. For these wildfire events, rapidfire provides good correlation with low error and bias, offering advantages over classical MLR or interpolation of the ground monitors alone. These results are similar to results from recent data fusion studies. Cleland et al. (2020) applied bias correction and data fusion methods to estimate PM2.5 impacts during the 2017 wine country wildfires with a resulting correlation of 0.71. They found that temporary monitors in the more rural areas were critical at improving results. Similarly, Zou et al. (2019) applied several machine learning approaches including random forest, to improve PM2.5 estimates across the Pacific Northwest (PNW) Aug-Sept 2017, with correlations ranging from 0.45 to 0.59. Note that the PNW region is much more sparsely populated with monitors than California.

Complete rapidfire results were also compared with available observations from the AirNow, IMPROVE, and CSN networks. Both IMPROVE and CSN collect 24-hr integrated filter-based measurements of speciated particulate matter every third day



**Table 5.** Performance metrics for four modeling methods

| Model | $R^2$ | RMSE | Median Bias | Normalized Bias | Median Error | Normalized Error |
|---|---|---|---|---|---|---|
| rapidfire | 0.74 | 21.5 | 0.083 | 0.76 | 2.13 | 18.6 |
| MLR | 0.68 | 23.8 | 0.056 | 0.49 | 2.59 | 22.6 |
| AirNow OK | 0.63 | 25.7 | 0.133 | 1.22 | 2.63 | 23.0 |
| PurpleAir OK | 0.38 | 33.3 | -0.095 | -1.04 | 3.75 | 32.8 |

(Solomon et al., 2014). IMPROVE PM$_{2.5}$ mass is determined gravimetrically. CSN no longer performs gravimetric mass analysis, but PM$_{2.5}$ is estimated by reconstructing total mass from the major components of PM$_{2.5}$: ammonium sulfate, ammonium nitrate, soil, organic matter, elemental carbon, and sea salt.

The comparison of AirNow data is shown in Figure 5. The vast majority of results are on along the 1:1 line. Concentrations
for 2018, 2020, and 2021 reach into the hundreds of micrograms per meter cubed, while concentrations in 2017 and 2019 are relatively lower. The model overestimates at the lowest concentrations and slightly underestimates the highest concentrations.

Figure 6 shows the CSN and IMPROVE monitor locations along with the identifiers used in this study. The rapidfire modeling shows excellent agreement with individual CSN and IMPROVE monitors as shown in Figure 7. This is somewhat surprising, as they represent a challenging test of the method. The 24-hr filter data are 100% independent of the model inputs and, for
IMPROVE especially, located far from other monitors in remote locations with complex terrain. Downsides of using these data are that the networks are sparser, sampling is only every third day, and the days with the highest concentrations are often not available as the IMPROVE sampler can clog in very heavy smoke situations.

The summary of results for the three monitoring network comparisons is shown in Table 6. As expected, AirNow performance statistics are very good, as these data were partially included in the model development. While CSN data were not used
in developing the model, the monitors are located in urban areas where they are close to the AirNow and PurpleAir monitors that were used in rapidfire. IMPROVE still shows good agreement, despite being far from observations used in the data fusions.

**Table 6.** Performance metrics for rapidfire at AirNow, IMPROVE, and CSN sites

| Network | $R^2$ | RMSE | Median Bias | Normalized Bias | Median Error | Normalized Error |
|---|---|---|---|---|---|---|
| AirNow | 0.90 | 6.80 | -0.09 | -0.99 | -0.01 | 18.5 |
| CSN | 0.87 | 5.18 | 0.42 | 3.93 | 1.96 | 15.3 |
| IMPROVE | 0.81 | 8.47 | 2.48 | 46.5 | 3.19 | 49.6 |

### 3.2 Characterizing rapidfire results across California

The results are plotted across California for two wildfire seasons: August - October, 2020 (Figure 8) and August - October, 2021 (Figure 9). In each case, daily average PM$_{2.5}$ reaches values greater than $200 \, \mu g \, m^{-3}$, with very strong spatial and temporal
variability. The 2020 case shows three widespread peaks, in August, September, and October. In the 2021 case, concentrations





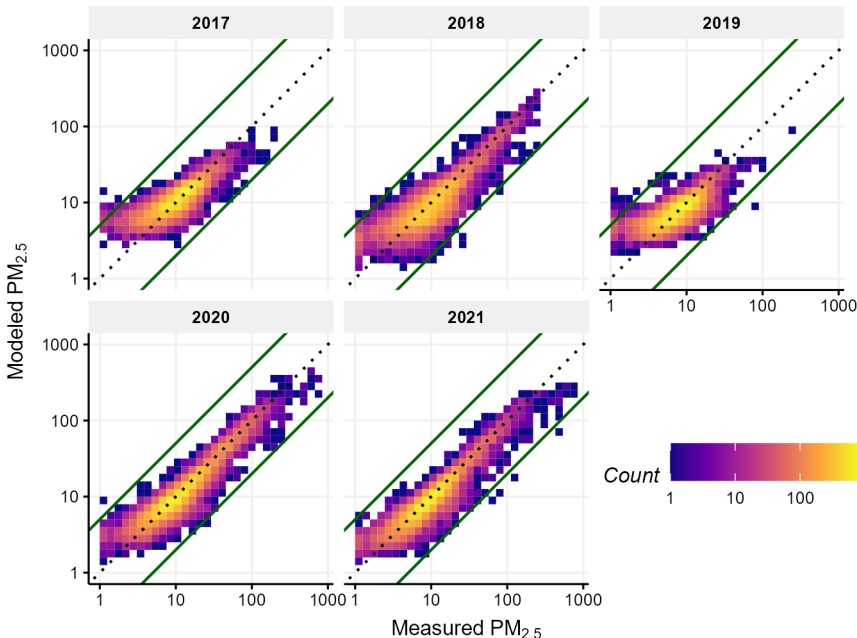

**Figure 5.** Model comparison against measured PM 2.5 from AirNow monitors.

were highest in northern locations in August, while values were higher further south in September and early October. These two cases highlight the complexity of these smoke events, which are controlled by multiple wildfires burning in and around the state simultaneously.

### 3.3 Excess mortality

As a demonstration of the utility of the rapidfire system, we adapted the methods of (Johnston et al., 2012) to estimate statewide mortality attributable to excess $PM_{2.5}$ during the wildfire seasons of 2017-2021. Excess mortality was estimated daily at the census tract level as:

$$Mortality\,attributable\,to\,PM_{2.5}\,exposure = \sum_{d=1}^{n} P \times M \times (PM_{2.5,d} - 15) \times RR_{SI} \tag{1}$$

where $PM_{2.5,d}$ is daily average $PM_{2.5}$ concentration predicted by rapidfire at census tract centroids, with minimum and
maximum values of 15 and 200 $\mu g\,m^{-3}$. Much of California has a relatively high baseline average $PM_{2.5}$ concentration during non-fire conditions. The minimum value of 15 was chosen to isolate the impacts of wildfire smoke on mortality, conservatively above typical daily mean $PM_{2.5}$ concentrations. Predictions were capped at 200 $\mu g\,m^{-3}$, as the $PM_{2.5}$ dose-response curve



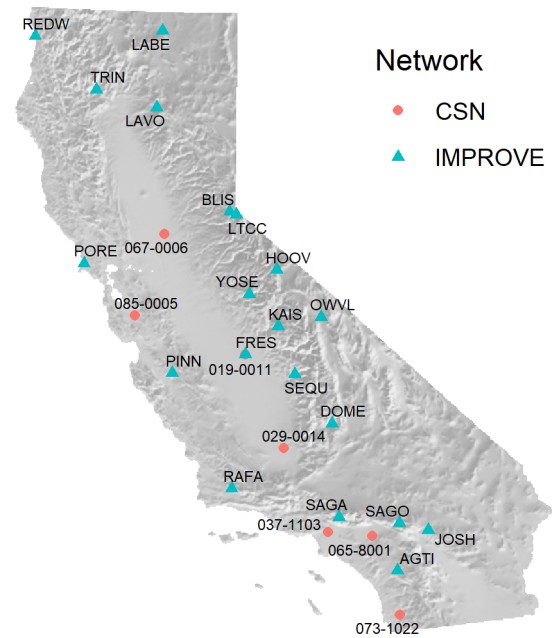

**Figure 6.** Map of CSN and IMPROVE monitoring stations used to validate model results.

flattens at higher exposures (Pope III et al., 2011). $M$ is the county-level, daily average mortality rate, which was acquired from the Centers for Disease Control's WONDER database (CDC, 2023), for the year 2016 (a recent, low-fire year). $P$ is the census tract population from the 2020 Census (Census, 2021). $RR_{SI}$ is the relative risk function for multiple-cause mortality due to short-term PM2.5 exposure. The value of $RR_{SI}$ was 0.11% per $1\,\mu g\,m^{-3}$ increase in PM$_{2.5}$ concentration (Johnston et al., 2012).

Figure 10 shows the California-wide daily excess mortality calculated from the increment of PM$_{2.5}$ concentrations above $15\,\mu g\,m^{-3}$. The most significant impacts are seen in 2018 and 2020. In November 2018, the Camp wildfire produced massive PM$_{2.5}$ emissions that transported throughout the Sacramento and San Joaquin Valleys and persisted under stagnant weather conditions. The nearly two-week period of high concentrations across a broad region of relatively high population density led to an estimated 266 excess deaths. The historic 2020 fire season was even more dramatic. Beginning in August, smoke from fires burning around the state contributed to an estimated 615 excess deaths across a three-month period. The spatial distribution of excess mortality for 2020 is shown in Figure 11. Impacts are shown by census tract. Though census tracts vary greatly in size, they have similar populations, with a minimum of 1,200 and and maximum of 8,000. Elevated excess mortality was widespread in the northern half of the state, especially away from the coast.





**Figure 7.** Model comparison against measured PM 2.5 at IMPROVE and CSN monitors

## 4 Discussion

### 4.1 Model input importance

Although the random forest model uses all of the provided predictor variables, the most explanatory variables are selected more often at each node. The relative importance of each variable can be visualized by calculating SHapley Additive exPlanations (SHAP) (Lundberg and Lee, 2017). SHAP quantifies the contribution of each predictor variable to the final model prediction. Figure 12 shows input values plotted versus SHAP for November 1-10, 2018. A single prediction, for CSN site 107-1001 on November 10, 2018, is highlighted. The SHAP values show the contributions to the final predicted concentration value from each of the model inputs. The individual component features of the model behave as expected from atmospheric dynamics. In





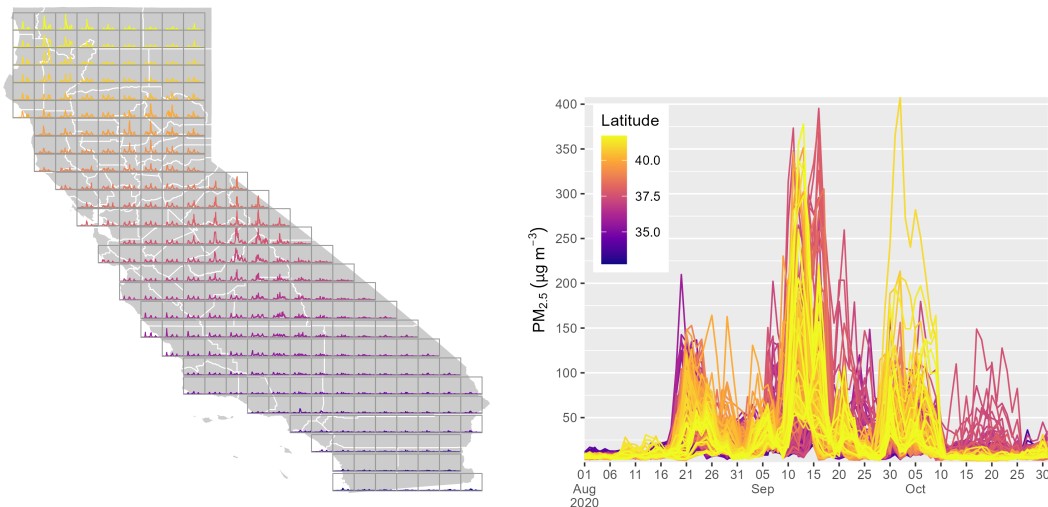

**Figure 8.** rapidfire PM2.5 estimates for August - October, 2020. Each box on the map shows the time series for a point at the centroid of the box and the larger plot shows all of those time series' overlaid.

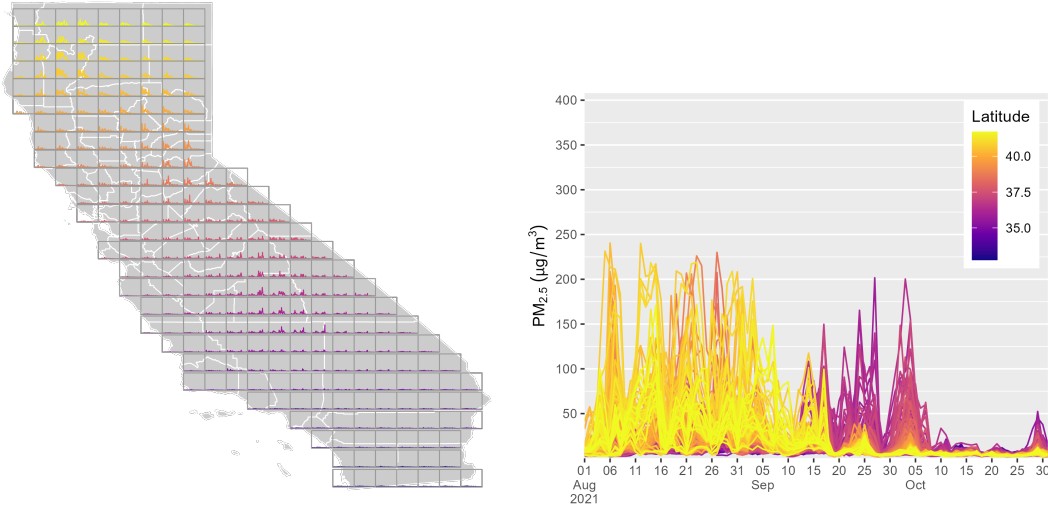

**Figure 9.** rapidfire PM2.5 estimates for August - October, 2021. Each box on the map shows the time series for a point at the centroid of the box and the larger plot shows all of those time series' overlaid.

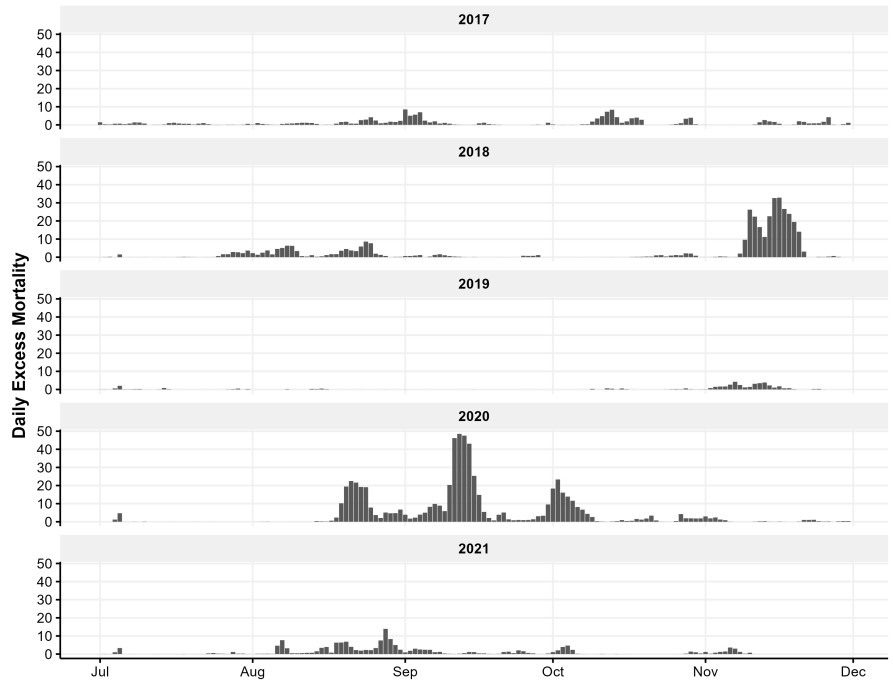

**Figure 10.** California-wide estimated daily excess mortality from $PM_{2.5}$ concentrations above $15\mu g\ m^{-3}$ for the period July-November, 2017-2021.

the highlighted case, PM$_{2.5}$ was high in the permanent and temporary monitors (Monitors), the sensor network (PurpleAir), and the smoke model (BlueSky). AOD was also elevated. By contrast, the planetary boundary layer (PBL Height) was low, as were wind speeds, humidity, and precipitation. Air temperature was moderate. The magnitude of the SHAP values in Figure 12 quantify the relative importance of the different inputs. The ground-base networks, both official monitoring and low-cost sensors are the most important variables in the model, followed by the BlueSky smoke model, planetary boundary height, and AOD. The remaining meteorological variables have a small, but coherent impact.

## 4.2 Application for health studies

The rapidfire modeling has been applied, and is being applied, in several epidemiological studies. The ability to produce wildfire-associated PM measures in a timely manner (about one month post event) allows time-critical planning and implementation for epidemiological studies. For example, when each of the recent large wildfires produced smoke plumes that covered urban areas of Northern California, the rapidfire modeling was used to determine the time periods and geographical areas where populations were most impacted by wildfire smoke. This information was used in two local studies, the Bio-Specimen Assessment of Fire Effects (B-SAFE) wildfire pregnancy cohort study and the WHAT-Now CA wildfire cohort study, to recruit participants from highly-affected areas to collect information and biological specimens to analyze later for wildfire-associated



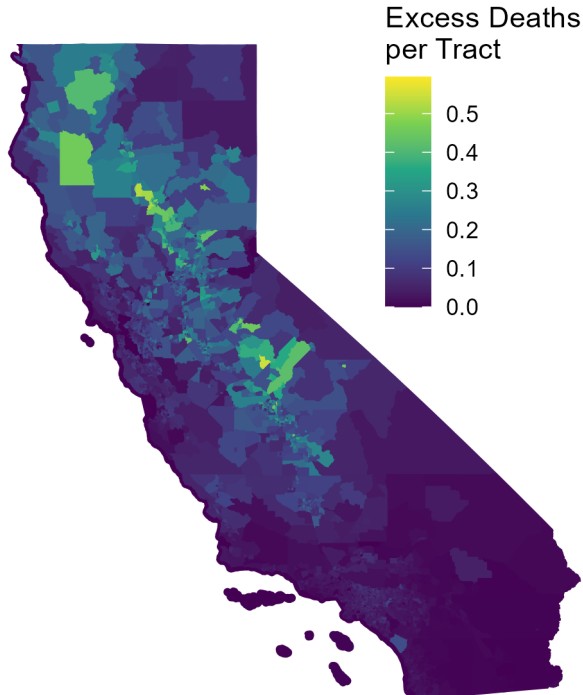

**Figure 11.** July-November 2020 excess mortality by census tract from $PM_{2.5}$ concentrations above $15\mu g\ m^{-3}$

compounds and biologic responses as indicators of potential for downstream health impacts. Both studies also related the

wildfire-associated $PM_{2.5}$ from rapidfire modeling to reported symptoms and health outcomes of the cohort participants. In B-SAFE, the timing and concentrations of $PM_{2.5}$ are being linked to birth outcomes of the children gestationally exposed to wildfires for the initial study, and in follow-up studies on respiratory, developmental, and other child conditions. Specimens collected in B-SAFE for those with higher versus lower modeled wildfire-associated $PM_{2.5}$ are also being compared across various measures (e.g., metals, contaminants, cytokines) to better understand differences by degree of exposure. In WHAT-

Now CA, $PM_{2.5}$ is being examined in association with respiratory outcomes. Both studies are planning to follow these exposed cohorts forward to examine later health outcomes.

Other local studies, including existing cohorts not focused on wildfire exposure, like the MARBLES (Markers of Autism Risk in Babies: Learning Early Signs) pregnancy cohort study of younger siblings of children with autism (Hertz-Picciotto et al., 2018), also used the rapidfire modeling in order to identify mothers and infants exposed to wildfire smoke while pregnant

and examine specimens being collected as part of the protocol for differences. Further, outcomes of these children, who are at higher risk of autism and other neurodevelopmental conditions, will be compared across those wildfire-exposed and unexposed.

Rapidfire modeling will be used to determine the time periods and geographical areas where populations were and will be





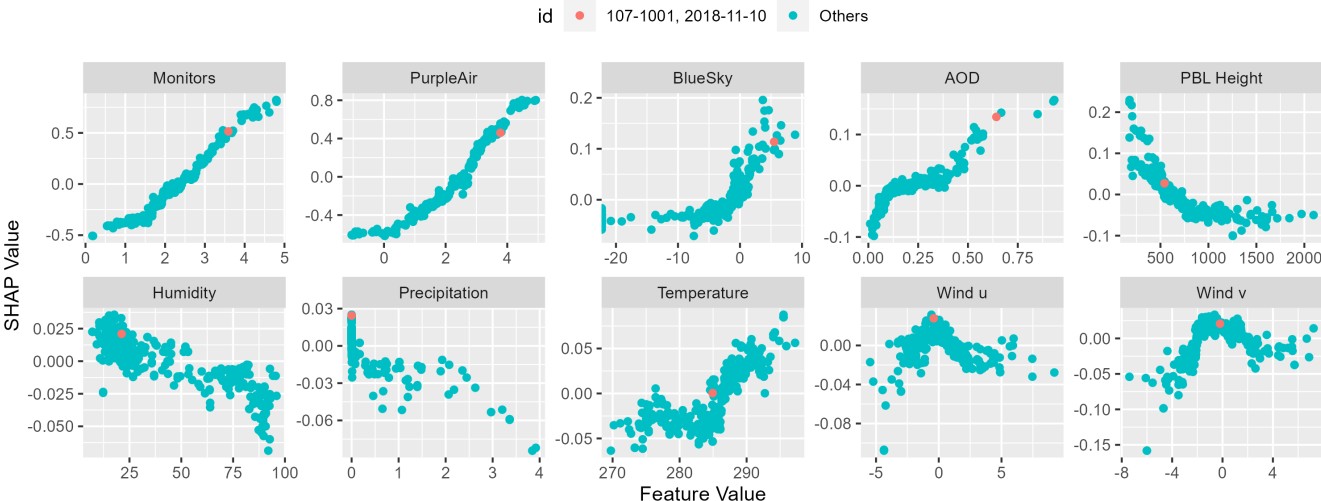

**Figure 12.** SHAP dependence plot at CSN and IMPROVE sites for November 1-10, 2018. Units for feature values depend on the variable and are listed in Table 3. BlueSky data were log-transformed in this plot for clarity.

most impacted by future wildfire smoke events for other statewide air pollution studies, including one funded by the EPA

(EPA STAR 84048401) that will link air pollution measures, including wildfire-specific air pollution, to birth outcomes and neurodevelopmental disorders, and work with the most affected communities to distribute education, materials, and tools for mitigating exposures.

### 4.3 Advantages over existing methods

There are many methods to produce spatially-resolved estimates of $PM_{2.5}$ for use in exposure studies. The advantages of

rapidfire include reliance on only off-the-shelf inputs with low latency, inclusion of data sets that provide improvements for wildland fire smoke, and an extensible framework with an open code base. If a new smoke event occurred, all inputs would be accessible and $PM_{2.5}$ modeling could be completed within one month. At present, only the NARR meteorological data is not available in near-real-time. In future work, this could be replaced by a daily operational model and the rapidfire predictions could be produced one day after an event. The addition of a low-cost sensor network has also significantly improved resulting

predictions.

### 4.4 Limitations

The rapidfire modeling approach has some limitations. The model requires high-quality training data to produce a high-quality result. In areas without accurate $PM_{2.5}$ measurements at point locations within the modeling domain, there is no way to create a reliable regression, though this is true for all statistical air quality models. In this study, the monitors from the AirNow network

served that purpose. However, AirNow is only present in the US, and the current rapidfire functions require data sets that are





not all globally available. These data sets could be replaced by others to cover a specific region, and new handling functions could be added to rapidfire to support those data sets as needed.

The rapidfire methods are designed with wildfire smoke events in mind. They are best suited for regional-scale modeling at spatial resolutions of 1-km or larger. This is appropriate for smoke events, which are driven by a regional source that impacts a broad swath. rapidfire would be less suitable for modeling exposure to PM from emission sources at very fine spatial scales, such as near-road emissions. Also, rapidfire is currently limited to estimates of total $PM_{2.5}$ only. Estimates of $PM_{2.5}$ composition are not supported with the currently available inputs, though this is an area of future work.

The random forests regression method has historically been seen as a black box, with potential for good prediction, but limited ability to provide insight into the drivers of the model prediction and the underlying physical phenomena. However, the advent of new metrics for explaining machine learning models, such as SHAP, makes these models more useful and transparent.

## 5 Conclusions

The rapidfire R package was developed to model relatively accurate particulate information derived from inputs retrieved easily. It incorporates off-the-shelf data sets that are produced operationally and with low latency (< 1 month) within a machine learning framework. rapidfire takes advantage of the recent burgeoning of low-costs sensors around the world, in addition to traditional air pollution data sources such as ground-based monitoring networks and satellite-derived aerosol products. The rapidfire code is available for use and contribution at https://github.com/raffscallion/rapidfire. We demonstrated rapidfire modeling for five recent wildfire seasons in California and validated results against fully independent filter-based measurements of $PM_{2.5}$. rapidfire showed excellent performance, predicting $PM_{2.5}$ under heavy smoke with high accuracy, even at remote and elevated sites. An example calculation of conservative excess mortality from high $PM_{2.5}$ exposure in California showed large impacts, including an estimated 615 excess deaths in California over a three month period of intense wildfire smoke in 2020. rapidfire $PM_{2.5}$ estimates are currently being used in several health effects studies in California. In the future, we hope to expand the methods to include data sets that are of even lower latency. At present, the input that becomes available the slowest is the NARR meteorology, which is available at the end of each month. There are several candidate meteorological data sources that are available daily, which would allow for next-day estimates of $PM_{2.5}$. These low-latency estimates would be useful for rapid deployment, recruitment, and sample collection in epidemiologic studies.

*Code and data availability.* The current version of rapidfire is available from the project website: https://github.com/raffscallion/rapidfire under the licence GPLv3. The exact version of the model used to produce the results used in this paper (v0.1.3) is archived on Zenodo (DOI: 10.5281/zenodo.7888562), as are input data and scripts to run the model and produce the plots for all the simulations presented in this paper (DOI: 10.5281/zenodo.7942846).



340 *Author contributions.* Sean Raffuse wrote the rapidfire package, performed analysis, and wrote the manuscript. Susan O'Neill provided BlueSky data, contributed text and editing to the manuscript, and advised throughout. Rebecca Schmidt led the studies that used rapidfire and contributed text to the manuscript.

*Competing interests.* The authors declare no competing interests.

*Acknowledgements.* This work was funded by a Joint Venture Agreement between The United States Department of Agriculture, Forest

345 Service and the University of California Davis (16-JV-11261987-091). IMPROVE is a collaborative association of state, tribal, and federal agencies, and international partners. US Environmental Protection Agency is the primary funding source, with contracting and research support from the National Park Service. The Air Quality Research Center at the University of California, Davis is the central analytical laboratory, with ion analysis provided by Research Triangle Institute, and carbon analysis provided by Desert Research Institute. Partial funding provided by the US Forest Service Pacific Northwest Research Station. We thank Dr. Yufei Zou for his prior work applying machine

350 learning to wildland fire and his helpful suggestions for this manuscript. We thank Jonathan Callahan for his AirSensor2 R package and help with processing purple air data. The views expressed in this publication are those of the authors and do not represent the policies or opinions of any U.S. government agency.



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
