# Peer review of "A model for rapid $PM_{2.5}$ exposure estimates in wildfire conditions using routinely-available data - rapidfire v0.1.3"

_EGUsphere, 2023_

## Author Response (AR1)

*Note to Editor*

*Due to unforeseen changes in my computer environment between the initial submission and these revisions, there were differences in the latexdiff that I was unable to resolve, along with small issues with the Copernicus template. I appreciate you patience. The latexdiff erroneously lists most of the citations as differences. Also, the final sections (code availability, contributions, acknowledgements) are missing their headings. I am hopeful that these can be resolved working with a latex expert on the editing team.*

The authors would like to thank the reviewers for their detailed, insightful, and relevant comments. They have identified important considerations and have improved the manuscript. Please see point-by-point responses to RC1 and RC2 below.

**RC 1**

*- The title and sections of the writeup refers to smoke-specific exposure but this paper focuses on total PM5 exposure, including wildland fire smoke. Recent work has provided estimates of smoke-specific PM2.5 exposure (e.g., Aguilera et al., 2023 and Childs et al., 2022). I recommend that the authors discuss whether smoke-specific exposures can be identified with their method. While the authors do attempt to remove effects of other PM2.5 sources in their health analysis, the choice of 15 μg/m3 is not adequately justified, given that significant spatial variability exists in non-fire smoke sources and concentrations across California.*

The reviewer is correct that the core methods presented are focused on total PM2.5 exposure. They were developed with wildfire conditions in mind, and tuned to those events, but ultimately provide total PM2.5. We agree with the reviewer that the title is misleading and suggest the following alternative. "A model for rapid PM2.5 exposure estimates in wildfire conditions using routinely-available data - rapidfire v0.1.3".

The title has been updated.

The health analysis is presented as a demonstration of utility; however, we agree that it can be made more useful by accounting for spatiotemporal variability of non-fire PM2.5 in California. Therefore, we have developed a conservative baseline PM2.5 by taking three recent lower fire activity years (2016, 2019, and 2022) and calculating the 90th percentile of daily PM2.5 by month and county based on AirNow monitors. Comparing these results to our more simplistic 15 ug/m3 across the board estimate, they are surprisingly similar. For the two time periods highlighted in the paper (November 2018 and August-October 2020), the 90th percentile method results in an estimated 254 and 652 excess deaths respectively, compared to 266 and 615 from the previous method.

The updated text is at line 245. Figures 10 and 11 are also updated to reflect the new calculation.

Aguilera et al., 2023 and Childs et al., 2022 both employ the NOAA Hazard Mapping System's (HMS) human-analyzed smoke plume extents to segregate days into smoke/non-smoke. While the HMS approach is reasonable, it is susceptible to false-positives when smoke is visible in satellite imagery but not present at ground level. In the future, we would like to explore the use of speciated monitoring data to better characterize and partition smoke-influenced days.

*- The abstract refers to studies employing multiple data sets that are costly. It may be helpful to the reader to include a discussion and specific references to such studies in the Introduction.*

The costly part of developing exposure estimates is in the labor required. A primary goal of this work is in developing tools that make updating exposure calculations for new cohorts and time periods a more automated process. Significant effort may be required to gather, format, and preprocess inputs. We have eliminated the word "expensive" from the abstract as it is generally redundant with "time consuming."

This is also in reference to chemical transport modeling, and specifically a response to our experience with modeling the 2017 Northern California wildfires as part of the NASA Health and Air Quality Applied Sciences Team (see O'Neill et al., 2021). In this case, methods were developed as part of a research project, but it is difficult to reproduce those methods for new time periods without tools. That is an issue we are trying to address with this suite of tools.

*- Table 1 lists Northern California fires, but Figure 1 shows all California fires. Why are Southern California fires not included in the table? It appears that the domain of the analysis and evaluation includes all of California, and this should be updated to be consistent. Relatedly, does the area burned represent Northern California or some other spatial domain? It may be more relevant to include the area burned for the fires studied in this work.*

Thank you. We corrected the caption and Table 1. The annual acres represent all of California and large fires from Southern California have been added to the table.

*- Table 2 – It would ease users in running the software to suggest including additional details in table 2 including the data type and resolution and key parameters for each rapidfire function.*

Thank you for the suggestion. We have added the spatial resolution to Table 2. For the key parameters for each rapidfire function, we refer the reader to the function documentation available in the package. (e.g., ?rapidfire::get_airsis_daterange)

*- L98 – Monitors included in AirNow encompass a range of measurement methods. It may be more accurate to compare performance of the EBAM to a specific monitor type. The authors should also consider addressing recent concerns about substantial bias in T640 monitoring instruments at high concentrations. Why are AirNow data used rather than data from EPA AQS, which have higher quality?*

The EBAM comparison studies used a BGI Inc. PQ-200 (Trent 2006) and Met One Instruments BAM (Schweizer et al. 2016). We added this information in the text at line 97. Then, the T640 monitors are optically based federal equivalent method (FEM) monitors and recent work of Long et al. (2023) highlight concerns with both positive and negative measurement artifacts when comparing T640 results with FRM filter based measurements. Investigating the California permanent monitoring network shows that there were approximately 2-9 T640 monitors in use in southern and eastern/central California depending on the study year. Otherwise, most permanent monitors (> 100) are either beta attenuation based or gravimetric. Also, in some cases purple air sensors were in the vicinity of these T640 monitors. One of the benefits of using machine learning with multiple datasets is ideally the capitalizing of strengths and reduction of issues associated with these disparate datasets. We agree though that if this is an issue with the T640's, their use as FEM's could be compromised during periods of high PM2.5 concentrations from

wildfire smoke. The rapidfire package can be easily modified to remove these monitors if a user desires; however, we felt their inclusion was beneficial because they provide critical information in locations of high aerosol loading where routine monitors do not exist.

Finally, the reason that AirNow is used as a data source, rather than AQS, which includes the finalized, human-quality-controlled data is data latency. Data are available in AirNow within hours, while it can be many months before the same time period is available in AQS. For most applications, AQS should be used; however, a goal of rapidfire is to have good PM2.5 estimates within one month. A reasonable and straightforward extension of the tools would be to develop a function to harvest AQS instead of AirNow for applications that are not time sensitive.

Russell W. Long, Shawn P. Urbanski, Emily Lincoln, Maribel Colón, Surender Kaushik, Jonathan D. Krug, Robert W. Vanderpool & Matthew S. Landis (2023) Summary of PM2.5 measurement artifacts associated with the Teledyne T640 PM Mass Monitor under controlled chamber experimental conditions using polydisperse ammonium sulfate aerosols and biomass smoke, Journal of the Air & Waste Management Association, 73:4, 295-312, DOI: 10.1080/10962247.2023.2171156

*- Low-cost sensor data – PurpleAir monitor data are routinely corrected (e.g., Barkjohn et al., 2022) to more closely agree with reference-grade monitors. The authors should consider applying a correction, or explain why this was not performed.*

We have elected to minimally correct our inputs, instead relying on the random forests model to capture interactions. The primary correction factor in Barkjohn et al. is relative humidity, which is included as a feature in our model. Similarly, we are not deriving a PM2.5 estimate from AOD but including boundary layer height and RH in our model. In other words, the value we use from PurpleAir is an indicator of PM2.5, and is not used as PM2.5 directly. We have added language in this section to address this (line 135).

*- L125 – The authors may want to address the recent switch to paid access to PurpleAir data by the provider.*

Thank you for the suggestion. We have added a note at line 130. The change in policy occurred as we were developing this manuscript and was incredibly frustrating.

*- L138 – While MAIAC is a good choice for this application, the authors should clarify that smoke plumes may also be mistaken for clouds in MAIAC (See for example Ye et al 2022). The authors may also wish to address how the rapidfire system can be updated in light of the upcoming end of life of the MODIS instruments.*

Thank you, this is a good point. All AOD algorithms can mistake thick smoke for clouds, and we have added language to this section (line 146).

The rapidfire system is designed to be modular. It would be straightforward to write new functions to deal with another AOD data source, such as VIIRS. We have added language to the discussion section to address this (line 301).

*- AOD gap filling – authors should test the effect of this imputation choice vs others (See e.g. Li et al., 2020) on the accuracy of the result. Consider including a binary flag in model training to denote pixels that are imputed. How were the imputation windows selected? How frequently are the different windows applied?*

Li et al., 2020 have done excellent work to produce high-quality, fully gap-filled AOD. However, the required effort to develop these is significant and thus does not fit with the goal of rapidfire, which is to produce good estimates of PM2.5 with off-the-shelf inputs that can be easily retrieved and used with minimal preprocessing. Even perfect AOD is an imperfect representation of ground-level PM2.5 during wildfire events.

A full assessment of the imputation method chosen is beyond the scope of this response (and the time provided to respond), but it is a good suggestion. We have added a section in the Limitations section on the improvements that can be made to assess and improve AOD imputation (as well as moving to MAIAC collection 6.1). This begins at line 328.

*- L183 – a large number of additional hyperparameters may be tuned for RandomForest. Which values were used for the other parameters, and how were the values selected?*

The random forest algorithm provides good results in default settings (Fernandez-Delgado et al., 2014) and is less tunable than other algorithms (Probst et al., 2018), which is one of its biggest advantages. Probst found that tuning the number of candidate variables at each split (i.e., mtry) provides the biggest improvement on average among possible hyperparameters. While there may be small gains available by further tuning other hyperparameters, we did not engage in that exercise. We used the default values for minimal node size, sampling scheme, and splitting rules.

Fernández-Delgado, M., Cernadas, E., Barro, S., & Amorim, D. (2014). Do we need hundreds of classifiers to solve real world classification problems?. The journal of machine learning research, 15(1), 3133-3181.

Probst, P., Boulesteix, A., & Bischl, B. (2018) Tunability: importance of hyperparameters of machine learning algorithms. Journal of machine learning research, 20(53).

*- L183 - Were interpolations rerun for each fold to exclude the hold-out monitors? Overall, more detail is needed on the cross-validation approach.*

Before developing the model, a random 30% of the monitor data was withheld as test data. Then, interpolations were created from the remaining 70% (training). The hyperparameter mtry was tuned using a 10-fold cross-validation on the training data. Those interpolations were not rerun for each subset. Once the model was developed, an additional 10-fold cross-validation was done to assess performance. For this exercise, interpolations were rerun for each fold, leaving out monitors in the test data set. For tests against independent data (e.g., IMPROVE and CSN), all monitor data was included and interpolated for use in predicting PM25.

We have reworked this model validation section (lines 197-218) for clarity.

*- Table 5 – The authors should provide additional detail about the alternate models used and how they were trained.*

In essence, each of the alternate models are subsets of presented model. Two are interpolation by ordinary kriging (AirNow monitors and PurpleAir sensors) using the same approach as used in developing the RF model. The final is a linear regression using the same inputs as the RF model. They were not trained but simply fit. Incidentally, we also tested other methods, such as XGBoost but did not find significant improvement over the RF model. It would be beneficial for others with machine learning expertise to further improve upon these methods.

*- L209 - Are these cross validation results or are the authors using the model training data for the comparison? Scatter plots should be shown for the cross-validation including interpolations with holdout set.*

This was erroneous. The results in Table 5 are now the correct cross validation results, while Table 6 uses the complete data set. The plot (Figure 5) has been updated to show cross-validation results, including interpolations with data withheld.

*- Table 6 and Figure 5 – If I understand correctly that these are used to show the results at locations and times where the model was trained, the authors should not show performance metrics or scatters for the AirNow monitors, since they will be impacted by overfitting and do not represent the performance of the model in a meaningful way. It would be appropriate to show addition detail of this nature for the cross validation, however. It is highly surprising that the performance for CSN and IMPROVE validation is better than the cross validation shown in table 5. Much additional detail is needed to understand how this could be possible, such as by addressing comments on the cross-validation approach mentioned above and by providing further details on the specifics of the data used in each step. The authors may wish to include descriptive statistics and/or scatter plots for the distribution of the data used in cross validation compared with that used in the independent tests. A 5-fold decrease in RMSE from cross validation (Table 5) to the independent test (Table 6) is difficult to believe.*

Thank you to the reviewer for highlighting some weaknesses in our approach. We have performed an updated validation exercise and present updated results. Figure 5 now presents the results of our cross-validation. We have removed the AirNow monitors from Table 6.

We redeveloped cross-validation results for Table 5 using a consistent time period and approach for each model. These updated results are more in line with the results of the final model versus CSN and IMPROVE shown in Table 6. The RMSE for IMPROVE/CSN are still lower than the cross-validation results; however, most of that difference can be explained by the lower dynamic range of the IMPROVE and CSN data. The highest values in the AirNow data are around 1000 ug/m3, while the highest values within the IMPROVE/CSN data are ~ 150 ug/m3. Artificially removing AirNow data with values above 150 ug/m3 results in a RMSE of 6, in line with CSN and IMPROVE. This is a limitation of the IMPROVE data, especially, as the sampler clogs at the highest concentrations.

*- Code: While I am not able to comprehensively bug check the code provided, it is important to note that I was unable to successfully install and run the software out of the box. Specific issues that should be addressed include: (1) automatically install dependencies from github as required instead of requiring the user to figure out how to do so, (2) dplyr, which is required for sample scripts test_processing.R, should be loaded in the script, and (3) bluesky_at_airnow is not exported in NAMESPACE but is called in test_processing.R. The authors may want to fully test the software in a new, clean environment to identify any other issues users might encounter.*

Thank you for your attempt and thank you especially for providing such specific feedback! We have made the necessary updates to the package and tested that it can be installed from a clean environment. The test_processing.R script was a vestige of earlier work and was removed.

*- Technical comments*

*- The use of PM, PM2.5, and PM5 throughout should be clarified. Are these meant to denote distinct concepts? They appear to be used interchangeably.*

This has been corrected.

*P1, L17 – increasing concentrations "due to wildfire smoke impacts."*

Thank you. Corrected.

*- P1, L19-20 – There seems to be a word missing in the sentence.*

Corrected.

*-L23 – A citation would be appropriate for the suspected several-fold increase in PM5 from smoke*

Thank you. Reference added.

*-L114 – Update the count for 2023 or update to past tense ("had grown"). Is this as of the beginning, end, or some other point of 2021?*

Updated and clarified.

*-Figure 4 – Authors should explain the meaning of "Fill 1", "Fill 2", and "Fill 3" in the figure legend or update the label.*

Clarified.

*-L200 – Are the authors suggesting that monitors are so dense that a basic interpolation method might work better? This statement seems unclear.*

This section has been rewritten.

**RC2**

*- Model uncertainty and "noise floor": The paper is generally rigorous in comparing rapidfire modeled PM2.5 to measurements (and those comparisons suggest good agreement), but more loose in its discussion and justification of how those modeled numbers were and can be translated to "smoke" and to the public health endpoints reported as examples in this paper. Is there a reasonable way to define an upper and lower bound for the rapid modeled concentrations? Is there a "floor" for those concentration numbers beyond which there's just model "noise"? This can be significant because even small concentration increments, applied over large populated areas in the downstream public health impact methods, can result in substantial excess deaths, so we want to make sure we keep model noise from adding to those death numbers to the extent possible. Above that "noise floor", how do we come up with*

*ranges for those otherwise impossibly precise numbers being reported? For example, was the range around the 615 excess deaths reported for that intense 3-month period during the summer of 2020 something on the order of 600-630, or more like 500-700? Further, how does that range change based on how good the agreement was between the rapidfire model and the other data sources? Even if there's no good answer for this at the moment, that could be more clearly articulated and discussed to strengthen the manuscript.*

Thank you for both your compliments and your insightful comments. We looked at r squared and normalized error as a function of PM2.5 concentration thresholds and found that the model begins to show skill at a threshold of about 10 ug/m3, with R^2 values less than 0.5 for data below that threshold. Normalized error reaches its floor at about 15 ug/m3. Thus, any value below about 10 ug/m3 is mostly model noise. As we are interested in smoke conditions, where concentrations varied from 20 - 1,000 ug/m3, we feel this is acceptable. However, the point that small increments can have large impacts on calculated excess deaths is well taken, and the uncertainty in the number of excess deaths includes, not only the PM2.5 estimate, but also the relative risk function, which is fixed in our study at 0.11% per ug/m3 increase in PM2.5. A full accounting of the uncertainties in the excess deaths estimate is beyond the scope of this paper, but we have added some bounds based on our RMSE. For the two intense periods mentioned in the paper, these produce ranges of 450-1070 excess deaths in the 2020 period and 200-340 in the November 2018 period. We have added some language discussing this at line 257.

*- Smoke vs. PM2.5: As RC1 notes, and related to the general "noise floor" issue above, 15ug/m^3 was selected as a threshold for distinguishing between smoke and PM2.5 in this paper's example, but background PM2.5 varies substantially in CA, higher for example in the Central Valley and metro areas, and much lower in many rural parts of the state. While it is understandable as a simplifying assumption to demonstrate the capabilities and potential of the rapidfire package, the authors could make it clearer that custom baselines may need to be used to get more realistic PM2.5 to smoke translation. They should make clear that these could substantially affect excess deaths, especially on the low end for areas with high populations. A sensitivity analysis would be the most robust way to address this issue. They might even consider providing an approach in the code for doing so, potentially based on an automated ingestion of existing spatial inventory data for non-fire PM2.5 sources.*

Thank you. Please see our response to RC1 above. The 15 ug/m3 threshold we selected was, in fact, quite conservative relative to typical concentrations for most of California for most time periods. However, you are correct that it is an underestimate for some locations and times. The slightly more precise but still conservative threshold outlined above (county-wide, 90th percentile monthly concentration) produced very similar results. We also produced a sensitivity analysis using our measurement error as explained above. Adding these sensitivity analyses into the code would be an excellent extension of the package, but the mortality calculations are not yet part of rapidfire.

*- Low-cost sensor correction: Please elaborate on whether the Purple air data was corrected as it is on the Fire and Smoke Map (https://fire.airnow.gov/), or if not, how you determined that wasn't important to do. Does the package include a way to do that?*

We have elected to minimally correct our inputs, instead relying on the random forests model to capture interactions. The primary correction factor in Barkjohn et al. is relative humidity, which is included as a feature in our model. Similarly, we are not deriving a PM2.5 estimate from AOD but including boundary layer height and RH in our model. In other words, the value we use from PurpleAir is an indicator of PM2.5, and is not used as PM2.5 directly. We have added language in this section to address this at line 135.

*- Code: Attempted to install the package and ran into the same issues as RC1 with dependencies and the code not being able to find the BlueSky files. As a result, this review could not include a replication of the reported results.*

Thank you for trying. We have updated the package in the hopes of addressing this.